# Large fault slip peaking at trench in the 2011 Tohoku-oki earthquake

Tianhaozhe Sun[1], Kelin Wang[1,2], Toshiya Fujiwara[3], Shuichi Kodaira[4] & Jiangheng He[2]

During the 2011 magnitude 9 Tohoku-oki earthquake, very large slip occurred on the shallowest part of the subduction megathrust. Quantitative information on the shallow slip is of critical importance to distinguishing between different rupture mechanics and understanding the generation of the ensuing devastating tsunami. However, the magnitude and distribution of the shallow slip are essentially unknown due primarily to the lack of near-trench constraints, as demonstrated by a compilation of 45 rupture models derived from a large range of data sets. To quantify the shallow slip, here we model high-resolution bathymetry differences before and after the earthquake across the trench axis. The slip is determined to be about 62 m over the most near-trench 40 km of the fault with a gentle increase towards the trench. This slip distribution indicates that dramatic net weakening or strengthening of the shallow fault did not occur during the Tohoku-oki earthquake.

[1] School of Earth and Ocean Sciences, University of Victoria, Victoria, British Columbia, Canada V8P 5C2. [2] Pacific Geoscience Centre, Geological Survey of Canada, Natural Resources Canada, 9860 West Saanich Road, Sidney, British Columbia, Canada V8L 4B2. [3] R&D Center for Earthquake and Tsunami (CEAT), Japan Agency for Marine-Earth Science and Technology (JAMSTEC), Natsushima-cho 2-15, Yokosuka 237-0061, Japan. [4] R&D CEAT, JAMSTEC, Showa-machi 3173-25, Kanazawa-ku, Yokohama 236-0001, Japan. Correspondence and requests for materials should be addressed to K.W. (email: kelin.wang@canada.ca).

The occurrence of very large slip on the shallowest part of the megathrust during the 2011 moment magnitude ($M_w$) 9.0 Tohoku-oki earthquake[1,2] is considered to be of paradigm-shifting importance in understanding tsunami generation and rupture mechanics[3]. Clear definition of the actual near-trench slip during the earthquake is critically needed for distinguishing between different trench-breaching slip scenarios that reflect fundamentally different fault mechanics[4]. If the most near-trench segment of the megathrust is not only an integral part of the seismogenic zone but also underwent the greatest stress drop (coseismic weakening), the resultant slip distribution should feature a large increase towards the trench. If the shallow segment strengthens with increasing slip rate (velocity-strengthening) but is unable to fully resist a large rupture propagated from the deeper seismogenic zone[5], the resultant slip distribution should feature a distinct decrease towards the trench. If the shallow megathrust exhibits velocity strengthening to resist slip at low slip rates but weakens to facilitate slip once a sufficiently high slip rate ($\sim 1\,\mathrm{m\,s^{-1}}$) is attained, a phenomenon known as dynamic weakening[6,7], the slip distribution may feature neither large increase nor large decrease towards the trench. All proposed models remain untested until we know the actual shallow slip.

However, despite the Tohoku-oki earthquake being by far the best instrumentally recorded subduction earthquake, the actual magnitude and distribution of the slip on the shallow megathrust are essentially unknown. A compilation of 45 published slip models, including those constrained by seafloor geodetic[8,9] and tsunami wave data[10,11] (Supplementary Tables 1 and 2), shows vastly different slip patterns in the most trenchward 100 km of the fault (Fig. 1 and Supplementary Fig. 1). The differences are due partly to various simplifications in inverting coseismic observations to determine fault slip. For example, many of the finite fault models, especially those used for inverting tsunami data, assume a planar fault and/or consist of rather large subfaults of rectangular shape[12]. Depending on how fault slip is constrained at the trench, the peak slip determined by the inversion may be located at the trench or some distance away from the trench. However, the primary reason for the poor state of knowledge is the lack of near-field observations of horizontal seafloor displacements: all seafloor global positioning system (GPS) measurements were made more than 50 km away from the trench[8,9].

Displacement observations nearest to the trench are the differential bathymetry measured before and after the Tohoku-oki earthquake by Japan Agency for Marine-Earth Science and Technology[1], which were not used by any of the 45 slip models in Fig. 1. By modelling these data using a finite-element deformation model, we are able to estimate the near-trench slip distribution along the main corridor (Fig. 1, inset). The inferred slip is of a very large average value ($> 60\,\mathrm{m}$) for the most seaward 40 km of the fault but with only a very small increase ($\sim 5\,\mathrm{m}$) over this distance, indicating neither large net weakening nor large net strengthening of this fault segment during the earthquake.

A SeaBeam 2112 with a frequency of 12 kHz and a beam width of $2° \times 2°$ was used to collect bathymetry data along track MY102 (Fig. 1, inset) in 1999, 2004 and in 2011 about 10 days after the earthquake[1]. Using the seaward side of the track as the reference, Fujiwara et al.[1] derived differential bathymetry of the landward side (Fig. 3d), hereafter referred to as the observed differential bathymetry (ODB). In deriving the ODB, only the inner beam soundings within a 45° swath width among the total available 120° swath width were used, because uncertainties in water sound speed affect the inner beam to a lesser degree than the outer beam. By cross correlating the bathymetries before and after the earthquake, Fujiwara et al.[1] estimated about 50 m horizontal and 10 m vertical motion of the landward side relative to the seaward side. This rough estimate did not invoke deformation modelling.

In this work, we model the 1999–2011 bathymetry differences for track MY102 reported by Fujiwara et al.[1] and quantitatively determine the near-trench slip in the main rupture area of the Tohoku-oki earthquake. The 2004 bathymetry data are not modelled except for testing purpose. The very short length of the seaward portion of the 2004 survey corridor, being only 1/5 of the 1999 survey, causes great difficulty in using it as the reference for ODB. Therefore the ODB based on the 2004 data is considered less reliable. Because seafloor displacement between 1999 and 2004 is expected to be very small, if not negligible, bathymetry differences between 1999 and 2004 provide an error estimate for ODB. The error thus estimated by Fujiwara et al.[1] in terms of inferred total horizontal seafloor displacement is about 20 m, or about 10 m if resolved to the trench normal direction. This error to a large part is due to the inadequate length of the seaward section of the 2004 data. When applied to the 1999–2011 ODB, the actual error should be much less but difficult to quantify. Nonetheless, we do not expect the 1999–2011 error in the trench-normal direction to be much larger than 5 m. Bathymetry data for track MY101 (ref. 2; Fig. 1, inset), with poorer quality than the MY102 data, will be discussed in the Discussion section.

Because the pre-event bathymetry data were collected 12 years before the Tohoku-oki earthquake and the post-event data were collected 10 days after, there is a question to what degree the deformation reflected in the OBD is truly coseismic. We think it is extremely unlikely that a large part of the 1999–2011 ODB could be due to fault creep before the earthquake. To produce tens of meters of coseismic slip, the shallow megathrust must have accumulated sufficient slip deficit prior to the 2011 earthquake, due to either actual fault locking by itself or the stress-shadowing effect of a locked patch immediately downdip. Large afterslip of the shallow megathrust in this area during the 10 days after the earthquake is also extremely unlikely, given the absence of interplate aftershocks along the main rupture zone which underwent large stress drop[14]. Modelling of post-seismic seafloor GPS measurements does not indicate afterslip in this area, although it does suggest large near-trench afterslip to the south of the main rupture zone[13].

## Results

### Differential bathymetry before and after the earthquake.
During the earthquake, seafloors on the two sides of the trench moved in opposite directions, with the motion of the landward side much larger than that of the seaward side[13]. For each point of the seafloor with fixed geographic coordinates (longitude and latitude), water depth is changed because of the motion and deformation of the sloping seafloor. The differential bathymetry is this change in water depth (Fig. 2).

### Synthetic differential bathymetry from deformation modelling.
We use a three-dimensional elastic finite element model that includes actual fault geometry and long-wavelength bathymetry (Supplementary Fig. 2). We add the model-predicted three-component coseismic displacements to the pre-earthquake bathymetry to produce synthetic differential bathymetry (SDB), in the manner illustrated in Fig. 2. This model allows us to study the role of internal deformation of the upper plate as well as its rigid-body translation along the megathrust in generating the ODB. For example, a trenchward decrease in slip causes

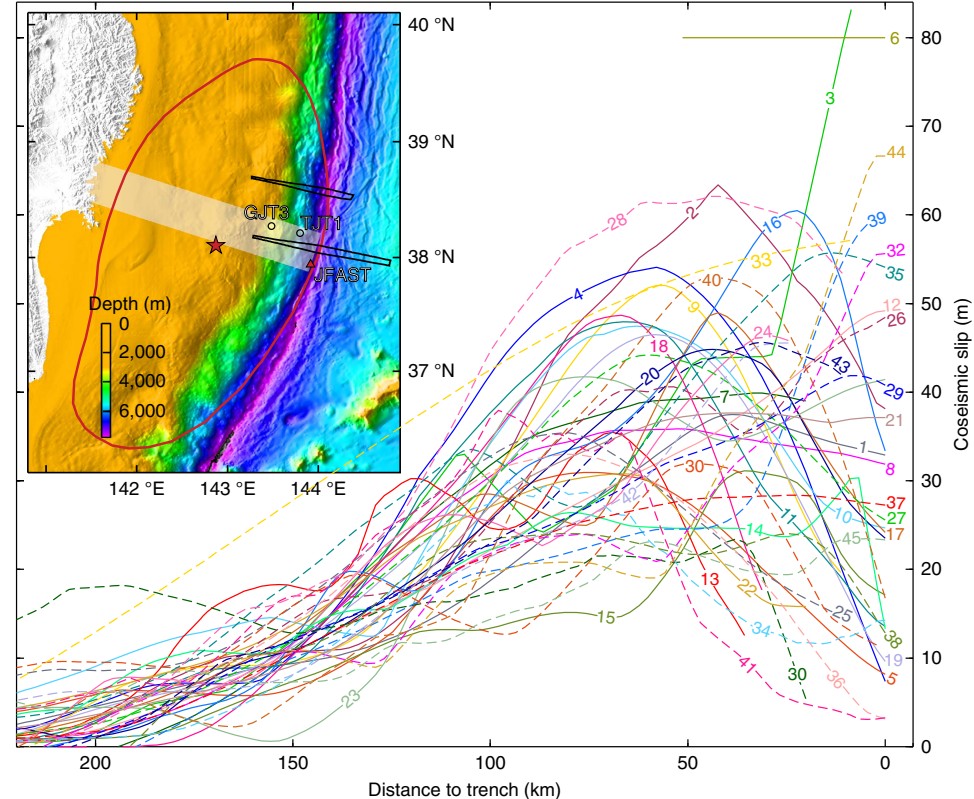

**Figure 1 | Compilation of 45 published slip models along a central corridor through the main rupture area.** The white band in inset shows the corridor. Each curve is labelled with its model number as in Supplementary Tables 1 and 2. Solid and dashed lines show models with and without, respectively, using seafloor GPS data as constraints. The subset of models that used tsunami data shows similar scatter of results near the trench (see Supplementary Fig. 1). In inset, red outline shows the 2-m contour of coseismic slip of the 2011 $M_w$ Tohoku-oki earthquake from ref. 29, and star shows the epicentre. The two differential bathymetry tracks studied in this work are outlined in black. Track MY102 along the central corridor is the main focus of this study. Track MY101, ∼50 km to the north, is discussed in Discussion. GJT3 is a nearby seafloor GPS site[9], and TJT1 is a nearby ocean bottom pressure (OBP) gauge site[20]. Red triangle shows the JFAST drilling location[16] where samples of the subduction fault zone were retrieved.

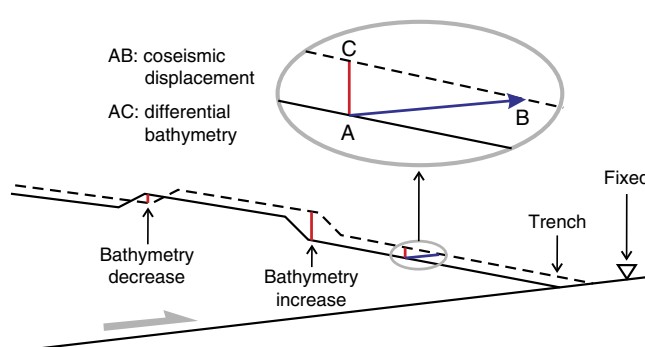

**Figure 2 | Cartoon illustration of the generation of differential bathymetry by a trench-breaching subduction earthquake.** Solid and dashed lines show the bathymetry before and after the earthquake, respectively. While coseismic deformation is of long wavelength, local seafloor slope variations can lead to coherent short-wavelength features of bathymetry decrease or increase.

horizontal shortening and enhances seafloor uplift, and vice versa. We have tested hundreds of SDB models. Those subsequently discussed in this article are summarized in Supplementary Table 3.

The SDB models for this small area are essentially two-dimensional, for lack of adequate constraints on along-strike variations of near-trench slip. For modelling convenience, we assign the same slip distribution over a wide along-strike

range (>400 km) that is much wider than the actual rupture area of the Tohoku-oki earthquake. Because the studied near-trench fault segment is quite shallow (<10 km below sea surface), seafloor displacement is sensitive mainly to the fault slip right beneath the track and very insensitive to assigned slip more than 20 km away to the north and south.

We determine three parameters by comparing our SDB models with the ODB. The first two parameters are the average slip and the slip gradient over the most near-trench 40 km of the megathrust. The third parameter, a depth adjustment to the SDB of the landward side, is to account for a remnant depth bias in the acoustically derived ODB. Although temporal and spatial variations in water temperature, especially at shallow depths (<2,000 m below sea surface), have been accounted for in deriving sound speed structure of ocean water for ODB determination, remaining uncertainties still lead to some remnant depth bias in the bathymetry data, even after maximizing cross correlations of the seaward (reference) side of different surveys. This is reflected in the ratio of the vertical to horizontal motion (∼10 to 50 m) of the landward-side seafloor relative to the seaward side estimated by Fujiwara *et al.*[1] This ratio would require a fault dip >10° near the trench that is much higher than the actual dip of ∼5°.

**Optimal slip model along the main track.** We search the model space defined by the three parameters described above to find the optimal SDB (Fig. 3c) that best matches the 1999–2011 ODB (Fig. 3d) and hence minimizes the root mean square

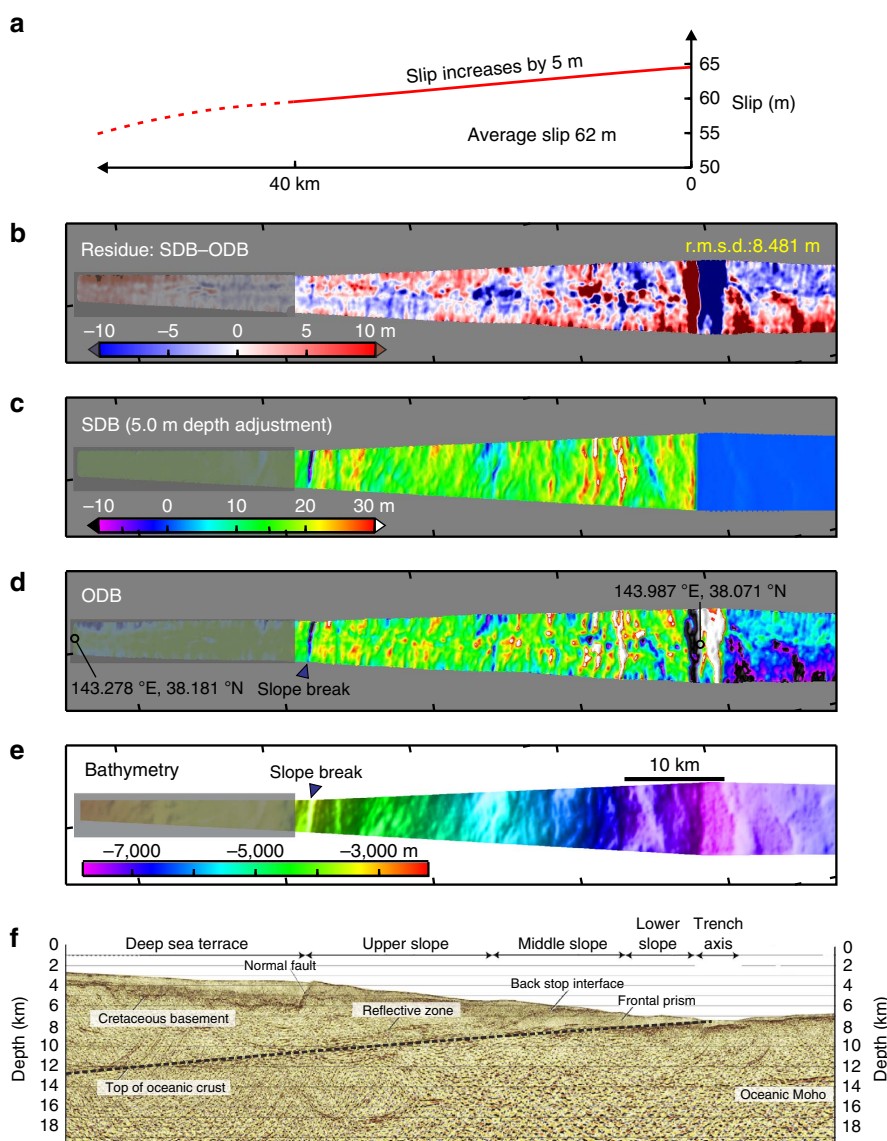

**Figure 3 | The optimal SDB model along the central corridor.** The location of the corridor is shown in Fig. 1 inset. (**a**) Fault slip distribution over the most seaward 40 km. (**b**) Residue between the SDB (**c**) and 1999–2011 ODB (**d**). (**c**) SDB produced using the slip distribution shown in **a**. (**d**) 1999–2011 ODB. (**e**) Bathymetry acquired in 1999. The deep sea terrace segment (<3,500 m; shadowed in **b**–**e**) has large uncertainties in water sound speed and the interpreted seafloor depth. Possible submarine landslides at the trench axis are not modelled in the SDB. Therefore, these segments as well as that seaward of the trench are not included for calculating the r.m.s.d. (**f**) Seismic reflection section along the same track[2]. Thick dashed line shows the megathrust fault.

deviation (r.m.s.d.) from the OBD (Fig. 4). We have also done the search by minimizing the mean absolute deviation of SDB from OBD and obtained the same results as with the r.m.s. Incoherent short-wavelength fluctuations in the ODB associated with sea and seafloor conditions, stability of the acoustic and navigation systems, and errors in local water temperature and salinity profiles[1] are not minimized, partly responsible for the relatively large r.m.s. For our study, the useful information is from long-wavelength coseismic deformation and coherent short-wavelength differential bathymetry due to topographic shift as shown in Fig. 2. The useful information is reflected in the r.m.s. differences between different models that are based on the same data set.

The optimal model for the main corridor (Fig. 3) requires an average fault slip of ∼62 m in the most seaward 40 km of the megathrust with the slip increasing towards the trench by 5 m over this distance. The resultant bathymetry change is due to a combination of the updip motion of the overriding plate

along the megathrust, seaward motion of the sloping seafloor and internal deformation of the upper plate. On the background of an overall uplift, coherent short-wavelength uplift and subsidence features are generated by local seafloor slope variations (Fig. 2), such as at the slope break between the deep sea terrace and the upper slope (Fig. 3e,f). Some of the differences between the optimal SDB and the ODB, especially in the amplitude of the short-wavelength features near the trench, may be due to inelastic deformation during or shortly after the earthquake[15] that are not modelled in this work. They also contribute to the relatively large r.m.s. In addition, for comparison, the optimal SDB based on the less reliable 2004–2011 ODB is shown in Supplementary Fig. 3.

In deriving the SDB, there is a trade-off between the average slip and the depth adjustment, as shown in Fig. 4 where the slip gradient is fixed at the optimal value of 5 m over the most seaward 40 km. For example, a model with a near-trench slip of ∼90 m but with no depth adjustment can also produce

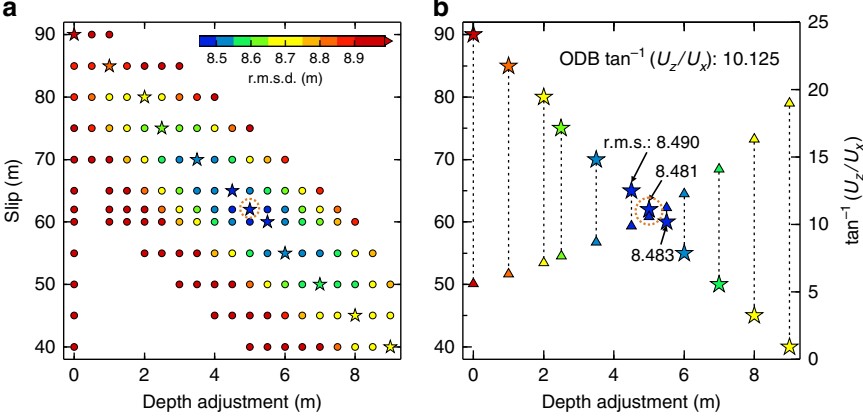

**Figure 4 | Search for optimal SDB in the parameter space for the central corridor.** (**a**) R.m.s.d.'s of SDB models as a function of average slip and depth adjustment. Slip gradient is fixed at the optimal value (5 m increase over 40 km) for all the models. Stars represent the best models (lowest r.m.s.d.'s) given slip value. The maximum r.m.s.d. is 12.5 m (for 40-m slip and 0-m depth adjustment), but the colour scale saturates at 8.9 m. (**b**) Ratio of average vertical to horizontal motion ($U_z/U_x$) of the seafloor as a function of slip magnitude and depth adjustment. Each star–triangle pair represents one model, with the star being the same as in **a** (same r.m.s. colour scale) and the triangle showing the corresponding arctangent value of ($U_z/U_x$). The $\tan^{-1}(U_z/U_x)$ of the optimal SDB agrees with the interpretation of the ODB ($\sim$10.125°) through cross correlation[1]. In both panels, the orange dashed circle marks the optimal SDB model shown in Fig. 3.

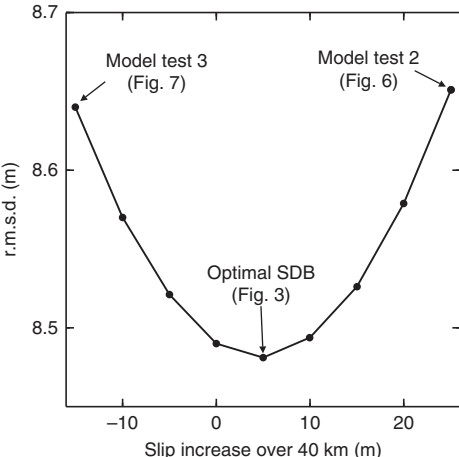

**Figure 5 | Sensitivity of SDB to slip gradient for the central corridor in terms of increase over the most near-trench 40 km.** Trenchward increase (as in Fig. 3a) is positive. Average slip is fixed at the optimal value of 62 m; the optimal depth adjustment varies with the slip gradient (not displayed).

$\sim$10–20 m water depth decrease as in the ODB, but the resultant seafloor displacement poorly explains short-wavelength features in the ODB and results in a larger r.m.s.d. (Supplementary Fig. 4). The 5 m adjustment for the remnant depth bias required by the optimal average slip 62 m accounts for the problematic fault dip ($>$10°) mentioned above (Fig. 4b).

**Slip gradient**. The frictional behaviour of the shallow megathrust during the earthquake is reflected not only in the magnitude of the slip but also in how the slip changes towards the trench. The sensitivity of SDB models to the slip gradient is illustrated by Fig. 5, where the average slip is fixed at the optimal 62 m. These tests indicate that the SDB is not very sensitive to small changes in the slip gradient, such that assuming 0 or 10 m increase (over 40 km) will not produce a very different SDB from using the optimal value of 5 m. However, using the SDB results, we can confidently reject some larger slip gradient values that are more diagnostic in reflecting fault frictional behaviour. For example, increasing (Fig. 6) or decreasing (Fig. 7) the gradient by

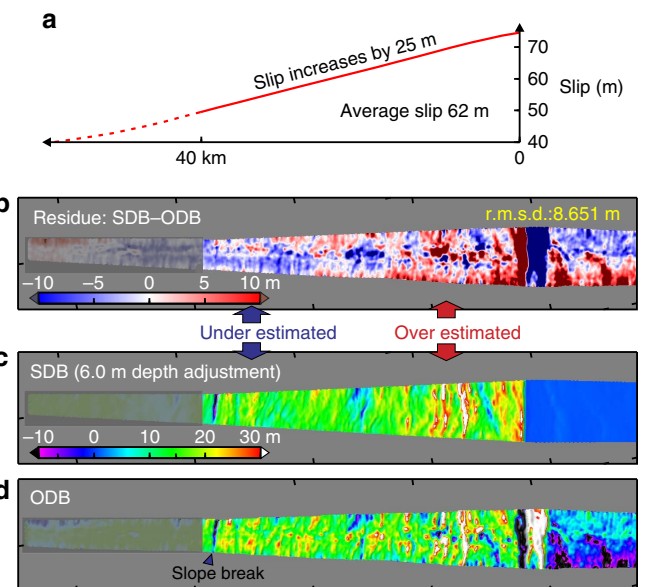

**Figure 6 | Testing SDB model for the central corridor with large trenchward increase in fault slip.** Otherwise the figure is similar to Fig. 3. (**a**) Fault slip distribution over the most seaward 40 km. (**b**) Residue between the SDB (**c**) and 1999–2011 ODB (**d**) showing overestimate of differential bathymetry near the trench but underestimate away from the trench. (**c**) SDB produced using the slip distribution shown in **a** and optimal depth adjustment 6.0 m. (**d**) 1999–2011 ODB.

20 m from the optimal value of 5 m over the most seaward 40 km obviously degrades the SDB's fit to the long-wavelength ODB. In other words, to explain the OBD in the main rupture area, the required coseismic slip exhibits neither large increase nor large decrease towards the trench.

## Discussion

The large ($>$60 m) slip with a gentle updip increase ($\sim$5 m) on the shallow megathrust shows a pattern different from nearly all the published rupture models in the main rupture area (Fig. 8a). Uncertainties in this slip distribution are reflected in the sensitivity plots of Figs 4 and 5. This result allows us to narrow

the range of possible slip behaviour scenarios as outlined in the opening paragraph. On the basis of the results shown in Fig. 6, we can reject the scenario that the shallowest megathrust underwent greater coseismic weakening than the deeper part,

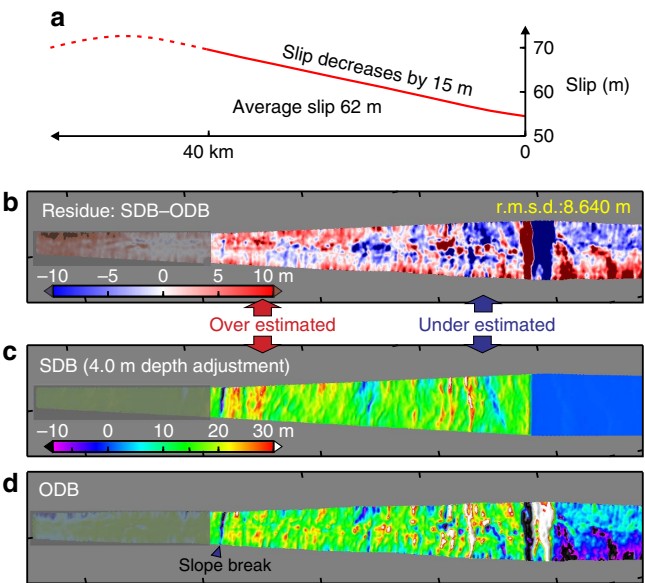

**Figure 7 | Testing SDB model for the central corridor with large trenchward decrease in fault slip.** Otherwise the figure is similar to Fig. 3. (**a**) Fault slip distribution over the most seaward 40 km. (**b**) Residue between the SDB (**c**) and 1999–2011 ODB (**d**) showing underestimate of differential bathymetry near the trench but overestimate away from the trench. (**c**) SDB produced using the slip distribution shown in **a** and optimal depth adjustment 4.0 m. (**d**) 1999–2011 ODB.

which would cause a large slip increase towards the trench and extreme stress drop on the shallowest megathrust (green curve in Fig. 8b). On the basis of the results shown in Fig. 7, we can also reject the scenario that the shallow megathrust persistently exhibited velocity strengthening during the rupture process, which would lead to slip decrease towards the trench and stress increase on the shallowest megathrust (red curve in Fig. 8b).

The optimal slip distribution (Fig. 3a) suggests that the shallowest segment of the megathrust along the central corridor must have weakened to a degree similar to the deeper epicentral area. This can be accomplished in two ways: (1) the shallowest segment shares the same frictional behaviour as the deeper seismogenic zone (blue curve 1 in Fig. 8b), or (2) the shallow segment exhibits velocity strengthening in the early phase of the rupture but dynamically weakens only when the slip accelerates to an adequately high rate ($>1\,\mathrm{m\,s}^{-1}$; ref. 6) (blue curve 2 in Fig. 8b).

Based on the information from drill core samples retrieved during the JFAST expedition from the shallowest part of the fault zone 7 km landward of the trench axis (ref. 16 and Fig. 1), the scenario represented by blue curve 2 in Fig. 8b is more likely. The core samples show both distributed (pervasive scaly fabrics) and localized (millimetre-scale slip zones) shear deformation within the plate boundary fault zone[16,17]. Co-existence of structures reflecting distinctly different modes of deformation is understood to imply rate-dependent frictional behaviour: the distributed deformation suggests low-rate velocity strengthening, while the localized slip zones may suggest high-rate ($>1\,\mathrm{m\,s}^{-1}$) dynamic weakening[17]. The rate-dependent behaviour is observed also in laboratory friction experiments on these core samples[18,19].

The ODB studied in this work allows us to determine shallow coseismic slip of the Tohoku-oki earthquake only in the main

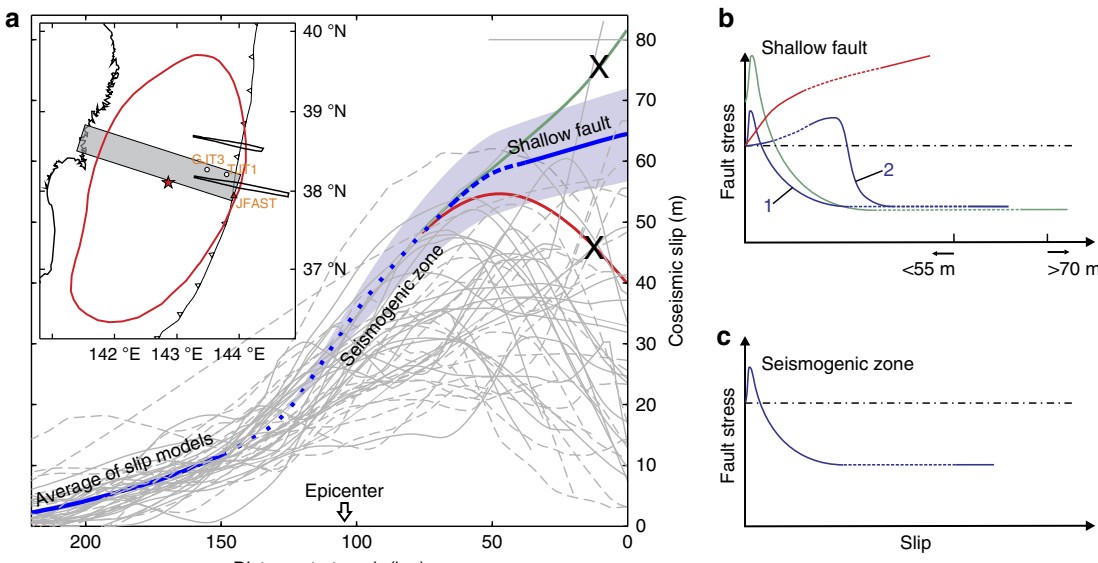

**Figure 8 | Illustrations of different mechanical behaviours of the shallow fault and their resultant slip distributions along the main corridor rejected and supported by SDB modelling.** (**a**) Comparison between the optimal shallow fault slip of this work (blue line) and the 45 published slip models shown in Fig. 1 (grey lines). The error range (blue shading) is based on models with r.m.s.d.'s $<8.55\,\mathrm{m}$ (Fig. 4a). Dotted part of the blue line is a hand-drawn, poorly constrained smooth connection between the near-trench slip determined in this work and the slip further downdip based on an average of the 45 slip models. Slip scenarios represented by the green and red lines are not supported by the SDB analysis. (**b**) Schematic illustration of stress evolution of the shallowest fault segment. Red, green and the two blue curves represent mechanically different shallow fault behaviours, corresponding to lines of same colours in **a**. Blue curve 2 represents a more likely scenario in which delayed dynamic weakening[6,30] of the shallow fault occurred during the earthquake. (**c**) Similar to **b** but for the deeper seismogenic zone.

rupture area (Fig. 8a). The slip must have varied along the Japan Trench as controlled by heterogeneous fault properties and stress conditions. For example, SDB modelling for bathymetry track MY101 (near 38.6° N), about 50 km north of our central corridor (Fig. 1, inset), shows a smaller average value ($\sim$40 m) but a larger increase (20 m) of slip to the trench, suggesting a higher degree of coseismic weakening of the shallow fault (Supplementary Fig. 5). The SDB results from both the central and northern tracks, together with the coseismic displacements recorded at nearby seafloor geodetic stations[8–9,20], can provide a much improved view of the trench-breaching slip of the Tohoku-oki earthquake as demonstrated by the slip distribution shown in Supplementary Fig. 6, which is obtained by hand-extrapolating the results shown in Fig. 3a and Supplementary Fig. 5a.

## Methods

**Deformation model.** We used the spherical-Earth finite-element code PGCviscl-3D developed by one of us (J.H.). The code uses 27-node isoparametric elements throughout the model domain. The effect of gravitation is incorporated using the stress-advection approach[21]. Coseismic rupture is simulated using the split-node method[22]. The code has been extensively benchmarked against analytical deformation solutions[23] and was applied to many subduction zone earthquake cycle modelling studies[24,25]. For modelling the coseismic deformation, the entire model domain is an elastic body. Other computer codes that can model elastic deformation, fault dislocation, and realistic fault and surface geometry will also suffice, although details of the model results could slightly differ if a Cartesian (as opposed to spherical) coordinate system is used and/or the effect of gravity is ignored or simulated in a different way. It can be readily shown that given slip distribution, the effect of spatial variations in rocks' mechanical properties on affecting elastic coseismic deformation directly above the thrust fault is negligibly small, although the effect can be larger for deformation farther away or if stress drop instead of slip distribution is prescribed to the fault. Therefore, we use uniform values for the rigidity (40 GPa), Poisson's ratio (0.25), and rock density (3,300 kg m$^{-3}$). We build a very large finite element mesh for the Japan Trench subduction zone to minimize the effect of the fixed lateral and bottom boundaries. The lateral boundaries are more than 1,000 km away from the rupture area, and the bottom boundary is set at 2,000 km depth (Supplementary Fig. 2). Subduction fault geometry is the same as in Sun et al.[25] and is constrained by earthquake relocation results and seismic reflection profiles[26–28], except that we have fine-tuned the dip of the shallowest part of the megathrust to 5° in accordance with the seismic imaging results in Kodaira et al.[2]

**Data availability.** All ODB data considered in this work have been published previously[1,2]. Other data such as those used to construct deformation models for SDB simulation are available on request from the authors.

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

## Acknowledgements

We thank the authors of the 45 published rupture models of the Tohoku-oki earthquake for providing the digital values of their slip models. T.S. was a member of the onboard Science Party of IODP Expedition 343 (JFAST). T.S. was supported by a University of Victoria PhD Fellowship, an Alexander and Helen Stafford MacCathy Muir Graduate Scholarship, a Bob Wright Graduate Scholarship and a Natural Sciences and Engineering Research Council of Canada discovery grant to K.W. This is Geological Survey of Canada contribution 20160230.

## Author contributions

K.W. and T.S. designed the study and did most of the writing. T.S. carried out most of the deformation and SDB modelling. T.F. and S.K. prepared ODB data and conducted data error analysis. J.H. wrote the computer code and participated in deformation modelling.

## Additional information

**Competing financial interests:** The authors declare no competing financial interests.

**Publisher's note**: 

