## [Peer Review File · Nature Communications]

Reviewer #1 (Remarks to the Author):

A. Summary of the key results

The presented manuscript "Large fault slip peaking at trench in the 2011 Tohoku-oki earthquake" presented by Sun et al. 2016, addresses and discusses a new data type namely "measured ocean acoustic profiles" in the use of inverse modelling. ODB data thereby is a very important datatype since ocean topography offers here information on the co-seismic deformation close to the trench where almost no direct observations are available.

B. Originality and interest: if not novel, please give references

The originality of the paper is limited because the manuscript does not clearly focus on the contribution of ODB data in inverse modelling. Fujiwara et al. already discussed all data related topics and drew conclusions on the near trench deformation. In addition, various other Authors have already shown that there has been peaking slip close to the trench for the 2011 Tohoku-Oki earthquake (see e.g. Hooper et al.). I see the scientific significance of the paper limited and do not recommend to publish the presented manuscript without major changes and severe improvements in the ODB data quality.

C. Data & methodology: validity of approach, quality of data, quality of presentation

The derived ODB profiles are used as input quantity for the inverse modelling but lack from a bias which may be present in the data coming from variations in water temperature over depths. The relation of acoustic parameters like pressure, temperature, saltiness etc. may not follow a linear relation to the measured profiles since already the geometry of the profiles follow a $1/\cos(\phi)$ relation. So the estimation of a single bias for the whole profile may be a weak simplification.

Do the residuals meet the criteria of Gaussian white noise? What noise properties characterize the ODB in the inverse modelling?

I am a little bit surprised by the large RMS fit of ~ 8.5 meter which I personally find poor. Looking at the maximal value of ~ 25 meter and the distribution of most of the data below ~ 10 meter in amplitude reveals a high uncertainty in the data or model. Moreover, the difference in the biases for different reference ODB is also quite high (~ 6 m). From the point of accurate seafloor modelling I would propose at least meter accuracy which is in agreement with DeSanto et al.. De Santo showed that the data of Fujiwara et al. have an accuracy of around 20 m which may not be appropriate for analysis.

A cross-track curvature of the measured deformation profiles at the deep sea terrace (Figure 2d) is clearly visible. It is very likely that the data is corrupted by instrumental errors. The model data (Figure 2c) show here flat terrain which seems appropriate.

The search in the parameter space Figure 3 shows only a small improvement of the model fit 8.9 to 8.5 ($\sim 5\%$ to "worst case"). Probably your "simulated" parameters are not very efficient. It seems that the model is not capable to model the fine scale ODB information properly.

The acoustic profiles offer an approximately resolution at 4 km depth of ~ 150 m and the model a resolution of around 5 km (what I can determine from the supplementary of Figure 3). How is this compatible? How does the mode treat the high resolution of the data and the involved spatial regularization of the solution?

In the supplement a brief statement about the rheological parameters is given. Is it true that you do not use any lateral nor changes with depth in the Poisson's ratio or rock density? When not true please show it.

From Supp. Figure 9, I do not see that the ODB data is consistent with the geodetic seafloor measurements. Already the same statement by Fujiwara et al. I found weak. Please compare the deformation rates for the closest points between the profiles and the geodetic measurements.

In general, when providing plots which show data outside the range of the colour bar please also indicate the end colours (see Figure 2d). In Figure 2b the colour code white means 0 and probably also values which are outside the colour range. This is quite misleading.

D. Appropriate use of statistics and treatment of uncertainties

Including ODB data in the inverse modelling would address a need for realistic error estimates of the data. Fujiwara et al. did not discuss these issues. I would like the Authors to contribute to this problem and to present an error assessment of the differential ODB data. Additional problems may rise which may introduce systematics for larger depths e.g. also given by the DOT (Dynamic Ocean Topography) which may be strongly present due to the Kuroshio current (Fuchs et al.).

The paper claims in the Abstract a proof of concept (dynamic weakening/strengthening). Can you give a level of confidence for this statement? Moreover, I miss an error estimate of your slip considering the error propagation of your measurements in the inverse modelling. Without this level of confidence, the statement of slip and the conclusions will always be vague.

Figure 1: I think it is not appropriate to compare all the different slip-models, since the models are based on different fault geometries, processing parameters like regularization, spatial scales, all based on various observation types where each different observation type may also show different spatial sensitivity and spatial resolution. I recommend to erase Figure 1 (or to put it in the appendix). Anyhow the different models do not contribute to the final conclusion made.

E. Conclusions: robustness, validity, reliability

I have severe doubts that high resolution ODB data, which only show a small fraction of the trench, can provide a proof of concept extrapolated on to a distributed fault slip model. The large RMS fit and the weak improvements searching the model space may emphasize this assumption. The Authors try to formulate an overall conclusion which is from my point of view not evidently given by the dataset.

F. Suggested improvements: experiments, data for possible revision

In your model approach the reference of your deformation is set on the whole area easterly of the trench. The ODB data would offer the very unique information about deformation very close and on the other side of the trench which never has been measured before and never (at least what I know) derived in any distributed fault-slip model. I think it would be of eminent importance to include this information in your modelling, choosing areas not affected by deformation as reference. This would incredibly increase the innovative potential of the manuscript and in addition highlight the potential of ODB-Data.

I suggest the Authors to compute a model with and without the ODB data and to directly show the impact of ODB data on the inverse modelling.

Moreover, an inversion with other datatypes like seafloor geodetic measurements and tsunami data is most important to evaluate the different data quality aspects.

H. Clarity and context: lucidity of abstract/summary, appropriateness of abstract, introduction and conclusions

The paper shows the methods and analysis applied, all written in a comprehensive and logical textual style with illustrative Figures and Plots.

Reviewer #2 (Remarks to the Author):

Wang et al evaluate the updip deformation caused by the 2011 Mw9.0 Japan megathrust earthquake looking at the difference between bathymetry profiles acquired before (mainly from year 1999) and after the rupture (year 2011). These data, already analysed in 2011 by Fujiwara et al. (2011), provided the first clear evidence that the megathrust rupture did reach the surface. Before the Tohoku earthquake, it had long been speculated that megathrust ruptures could not propagate through the shallow unconsolidated sediments. These bathymetry data proved otherwise and even demonstrated that the rupture can propagate to the surface with very large slip amplitudes (~60m). The aim of the Fujiwara et al. (2011) study was to release the result quickly. It was published as a Science brevia and therefore didn't go into the details of the analysis nor proposed a full discussion of the results. In this respect, the study proposed by Wang et al. addresses this lack and is very valuable.

The main scientific question posed by Wang et al. is whether a finer processing allows to distinguish a trend in coseismic slip amplitude (increase, stabilization or a decrease) towards the trench. This piece of information is critical to constrain the frictional properties of the shallow fault interface and its ability to accumulate stresses and/or release them during a coseismic rupture. The question is therefore of fundamental interest.

The manuscript is easy to read and contains appropriate references. Yet, I'm not convinced by the demonstration and the argumentation. Although I consider that this work represent a significant contribution, I feel that my concerns would be better address by a re-submission.

I list below my main concerns:

***despite significant efforts (finite-element modelling, grid search of optimal parameters, etc), the result of the analysis is not significantly different from the more simple and straightforward analysis of Fujiwara et al. (2011). Authors state that their new analysis allows to discriminate between scenarii of slip amplitude significantly changing toward the trench, but this was already suggested by the 2011 results: the residual of the differential bathymetry didn't show any clear gradient in the residuals. The previous study was aimed at quickly releasing the results and didn't push the interpretation: in that respect the present study brings a clear plus. The discussion is of general interest and needs to be published.

***several other bathymetry datasets are analysed in the supplementary section. These analyses reveal variable results, sometimes contradicting the main conclusion of the paper. This part is not sufficiently detailed considering that it shows results not in line with the main text. Two cases in particular:

*the 2004-2011 differential bathymetry (the main text is based on 1999-2011 profiles) is assumed unreliable because of a limited coverage seaward of the trench (assumed stable and used as a reference to calibrate the analysis). There is no clear explanation as to why 10km of seaward data is not sufficient to calibrate a 40km section: the residuals are small and do not show any suspicious trend (Fig. S7). This result is quite different from the 1999-2011 analysis: 12m decrease in slip for 2004-2011 versus 5m increase for 1999-2011.

*The authors also present in the supplementary section differential bathymetry results from a profile further north (Fig. S8). The results show a significant 20m increase of slip towards the trench suggesting that the shallow megathrust underwent coseismic weakening. This result suggest that there is not one single type of frictional response as proposed in the main text, but some along-strike variability.

***The authors motivate their analysis stating that finite-fault slip models of the Tohoku earthquake did not manage to resolve slip on the shallowest part of the subduction interface. I think this discussion is, for the most part, too long and not constructive:

*most finite-fault slip model listed do not pretend to resolve slip at the scale of the bathymetry profiles. For instance, Fujii et al. (2011) resolve slip on squares of 25 and 50km, bigger than the length of the bathymetry profile. Most studies listed are based on a planar fault geometry which usually fall km, sometimes more than 10, away from the trench.

*recent studies have highlighted the need to simultaneously incorporate tsunami and the 3D slab geometry to properly infer the details of the shallow slip (Bletery et al., 2014; Melgar and Bock, 2015). Even the models doing so have a spatial resolution of several km along strike, and show relatively strong variability near the trench. A fair and meaningful comparison should be limited to these recent studies and acknowledge their resolution and variability.

The final statement that "contrary to a common belief, tsunami observations do not necessarily provide resolution in determining near-trench slip" needs to be adjusted. Simple synthetic tests (e.g. Yokota et al., 2011; Romano et al., 2012; Bletery et al., 2014) reveal that tsunami data should be very good at resolving near-trench slip. The statement made by the authors could be true if tsunami data were not modelled properly (e.g. over-simplified fault geometry as in many studies) or if the authors are talking about length scales smaller than what is resolved by tsunami data (10km?). In the latter case, it is not obvious that differential bathymetry is doing better.

*** The argument that "Large afterslip of the shallow megathrust [...] is also extremely unlikely, given the absence of interplate aftershocks along the main rupture zone" is not supported by any observation. Indeed, several recent postseismic studies (Perfettini and Avouac, 2014; Yamagiwa et al., 2015) argue that the shallowest part of the megathrust

Additional comments and corrections:

_Figure 1 and S1. Romano et al. (2012), line 15, have a maximum amplitude of more than 30m (over 40m?). I leave the authors the responsibility to double check all profiles.

_a discussion of the shortening rates and the limit of elasticity could be a nice addition.

_a direct comparison with the result of Ito et al. (2011)

_data to reproduce the results are not included. The authors comment in their manuscript that no finite-fault source inversion study has included the results of the previous differential bathymetry analysis (Fujiwara et al., 2011): for this not to happen again, I urge the authors to provide their new results in ascii format in the supplements (coordinates of the profiles, slip amplitudes inferred).

References :

Bletery, Q., Sladen, A., Delouis, B., Vallée, M., Nocquet, J. M., Rolland, L., & Jiang, J. (2014). A detailed source model for the Mw9. 0 Tohoku-Oki earthquake reconciling geodesy, seismology, and tsunami records. *Journal of Geophysical Research: Solid Earth*, 119(10), 7636-7653.

Fujiwara, T., Kodaira, S., No, T., Kaiho, Y., Takahashi, N., & Kaneda, Y. (2011). The 2011 Tohoku-Oki earthquake: Displacement reaching the trench axis. *Science*, 334(6060), 1240-1240.

Ito, Y., Tsuji, T., Osada, Y., Kido, M., Inazu, D., Hayashi, Y., ... & Fujimoto, H. (2011). Frontal wedge deformation near the source region of the 2011 Tohoku-Oki earthquake. *Geophysical Research Letters*, 38(7).

Melgar, D., & Bock, Y. (2015). Kinematic earthquake source inversion and tsunami runup prediction with regional geophysical data. *Journal of Geophysical Research: Solid Earth*, 120(5), 3324-3349.

Perfettini, H., & Avouac, J. P. (2014). The seismic cycle in the area of the 2011 Mw9.0 Tohoku-Oki earthquake. *Journal of Geophysical Research: Solid Earth*, 119(5), 4469-4515.

Romano, F., Piatanesi, A., Lorito, S., D'Agostino, N., Hirata, K., Atzori, S., ... & Cocco, M. (2012). Clues from joint inversion of tsunami and geodetic data of the 2011 Tohoku-oki earthquake. *Scientific reports*, 2.

Yamagiwa, S., Miyazaki, S. I., Hirahara, K., & Fukahata, Y. (2015). Afterslip and viscoelastic relaxation following the 2011 Tohoku-oki earthquake (Mw9.0) inferred from inland GPS and seafloor GPS/Acoustic data. *Geophysical Research Letters*, 42(1), 66-73.

Yokota, Y., Koketsu, K., Fujii, Y., Satake, K., Sakai, S. I., Shinohara, M., & Kanazawa, T. (2011). Joint inversion of strong motion, teleseismic, geodetic, and tsunami datasets for the rupture process of the 2011 Tohoku earthquake. *Geophysical Research Letters*, 38(7).

Reviewer #3 (Remarks to the Author):

One of the most important lessons of the great 2011 Tohoku-oki earthquake is the unexpectedly large slip that seems to have occurred at shallow depth towards the trench. In the classical literature, the area between the trench and some up-dip limit of the seismic friction plane was suspected to be "weak" and creeping, in relation with the expected large amount of water carried by sediments at décollement-level.

This paper specifically addresses the rate of change of slip towards the trench axis, using bathymetric data that have been collected before and just after the earthquake. The data were already presented shortly after the event, but their exploration was never pushed to the level presented here. Beyond the general problem of the amount of co-seismic sliding towards the trench, the aim here is to reach the rate of change of this slip, since this value is expected to change drastically with mechanical behaviour. This approach is definitely novel.

The paper is very well written and presented, and I do not have much to say about the overall writings, illustrations, and references. One detail in fig 4b: curves labelled 1 and 2 (blue) are described in the text only, it could be useful to add a short text in the legend of the figure as well, and in the meantime, indicate that 2 is your favoured scenario.

Here are a few comments (dealing mainly with minor precisions on the methodology):

- Figure 1 is impressive, and the authors made a nice and very useful job in compiling these various models. There are clearly two classes of models in the 0 to 80 km range from the trench: some of the models show a bell curve (including or not including seafloor geodesy), while some others (few) show steadily rate increase. Is there any "rule" behind this (type of inversion used),

or is it that the behaviour towards the trench is just not constrained in most cases?

- Some of the models shown use a Bayesian approach, and it may be fair to state that in that case, the inversion does not lead to one single slip model but to a population of models. If the Bayesian approach is correct, then the solution proposed in this manuscript should belong to this population. I understand that it would be hard to check.

- The critical range used here covers the first 40 km of the wedge away from the trench. In Fujiwara et al. 2011 (using these spectacular bathymetric data for the first time), the area near the trench axis was excluded in relation with possible landslides. Perhaps a word should be added on the possible limitation that co-seismic sediment mobilization would produce. I guess that the area mentioned as landslide in their figure 1 has been excluded from the calculation (shows also in your figure 2d).

- The trade-off between the average slip and the depth adjustment parameter is well explained as an acoustic bias. Theoretically, it would be possible to link the final value of this depth adjustment to the kind of seasonal variations that are expected to be found in the water column, although it is most probably within the error bars of anything that could be calculated. Practically, there might be a way to check the water velocity law variation in the trench area, if a flat area exists: beams with larger aperture will wrongly bend upward or downward if the velocity law is incorrect, and comparing the required velocity laws before and after would ultimately lead to the adjustment. At least the sign of it may be retrieved ... not for this paper anyway!

- The final model proposed here is finally a sort of upper-bounding envelope of the models proposed so far. If this applies not only to this section, but also to the entire sliding area, would not the modelled magnitude of the earthquake be too large compared to the observed one?

My conclusion is that the paper is based on high-quality data, the methodology is robust and very well explained. The paper brings a new and important piece to the understanding of the co-seismic mechanical behavior during megathrust earthquakes.

Responses to Reviewers' Comments

Reviewer #1 (Remarks to the Author):

A. Summary of the key results

The presented manuscript "Large fault slip peaking at trench in the 2011 Tohoku-oki earthquake" presented by Sun et al. 2016, addresses and discusses a new data type namely "measured ocean acoustic profiles" in the use of inverse modelling. ODB data thereby is a very important datatype since ocean topography offers here information on the co-seismic deformation close to the trench where almost no direct observations are available.

B. Originality and interest: if not novel, please give references

The originality of the paper is limited because the manuscript does not clearly focus on the contribution of ODB data in inverse modelling. Fujiwara et al. already discussed all data related topics and drew conclusions on the near trench deformation. In addition, various other Authors have already shown that there has been peaking slip close to the trench for the 2011 Tohoku-Oki earthquake (see e.g. Hooper et al.). I see the scientific significance of the paper limited and do not recommend to publish the presented manuscript without major changes and severe improvements in the ODB data quality.

We respectfully disagree with the reviewer's opinion on the originality of this work. As is obvious to the other two reviewers, our work is clearly focused on the contribution of ODB data to deciphering coseismic slip of the shallow part of the megathrust and has provided new and quantitative information.

C. Data & methodology: validity of approach, quality of data, quality of presentation

The derived ODB profiles are used as input quantity for the inverse modelling but lack from a bias which may be present in the data coming from variations in water temperature over depths. The relation of acoustic parameters like pressure, temperature, saltiness etc. may not follow a linear relation to the measured profiles since already the geometry of the profiles follow a $1/\cos(\phi)$ relation. So the estimation of a single bias for the whole profile may be a weak simplification.

We did not make it clear that in processing the bathymetry data, we already considered the seasonal variations in water temperature over depths. For surveys at different times, Fujiwara et al. (2011) used different temperature-depth profiles to derive water sound speeds and the seafloor depths. We have revised the text as follows to clarify this:

“Although temporal and spatial variations in water temperature, especially at shallow depths (< 2000 m below sea surface), have been accounted for in deriving sound speed structure of ocean water for ODB determination, remaining uncertainties still lead to some remnant depth bias in the bathymetry data, even after maximizing cross correlations of the seaward (reference) side of different surveys.”

Do the residuals meet the criteria of Gaussian white noise? What noise properties characterize the ODB in the inverse modelling?

If we correctly get the message behind these questions, they are asking whether it is appropriate to use RMS (based on the L2-norm) as a measure of SDB – OBD differences. We actually had tested the mean absolute deviations (based on the L1-norm, much more tolerant to “outliers”) and obtained exactly the same optimal parameter values as with the RMS. This shows that the results are not sensitively dependent on the type of “noise”. We have clarified this by adding the following sentence after the first mentioning of RMS:

“We have also done the search by minimizing the mean absolute deviation of SDB from OBD and obtained the same results as with the RMS.”

I am a little bit surprised by the large RMS fit of ~8.5 meter which I personally find poor. Looking at the maximal value of ~25meter and the distribution of most of the data below ~10 meter in amplitude reveals a high uncertainty in the data or model. Moreover, the difference in the biases for different reference ODB is also quite high (~6m). From the point of accurate seafloor modelling I would propose at least meter accuracy which is in agreement with DeSanto et al.. De Santo showed that the data of Fujiwara et al. have an accuracy of around 20m which may not be appropriate for analysis.

The reviewer is misled in this regard, perhaps because we did not explain it adequately. We have added Fig. 2 (from previous Supplementary Information) to help the reader understand the issue. We have also added the following text right after the above newly added sentence:

“Incoherent short-wavelength fluctuations in the ODB associated with sea and seafloor conditions, stability of the acoustic and navigation systems, and errors in local water temperature and salinity profiles¹ are not minimized, partly responsible for the relatively large RMS. For our study, the useful information is from long-wavelength coseismic deformation and coherent short-wavelength differential bathymetry due to topographic shift as shown in Fig. 2. The useful information is reflected in the RMS differences between different models that are based on the same data set.”

We do not understand the reviewer’s comment about differences in depth biases between different models being large (“high”). We searched a wide range of depth bias values. Of course the values are very different between models.

A cross-track curvature of the measured deformation profiles at the deep sea terrace (Figure 2d) is clearly visible. It is very likely that the data is corrupted by instrumental errors. The model data (Figure 2c) show here flat terrain which seems appropriate.

We already pointed out this issue in the figure caption (and shaded that area) which

may have escaped the reviewer's attention. Because of the large uncertainties, we excluded this area from RMS calculation. Our effort is devoted to overcoming such difficulties to extract valuable information from these data. We hope the review will agree with us that we should not "throw the baby out with the bath water."

The search in the parameter space Figure 3 shows only a small improvement of the model fit 8.9 to 8.5 (~5% to "worst case"). Probably your "simulated" parameters are not very efficient. It seems that the model is not capable to model the fine scale ODB information properly.

The percentage comparison (5%) is logically incorrect; see reply to the "I am a little bit surprised ..." comment above. However, the reviewer's comment does remind us that the color bar in the original figure is confusing. We have modified it to show that the actual upper-bound is much higher than 8.9. We have also added the following sentence to Fig. 4 (previous Fig. 3) caption:

"The maximum RMS deviation is 12.5 m (for 40-m slip and 0-m depth adjustment), but the color scale saturates at 8.9 m."

The acoustic profiles offer an approximately resolution at 4km depth of ~150m and the model a resolution of around 5km (what I can determine from the supplementary of Figure 3). How is this compatible? How does the model treat the high resolution of the data and the involved spatial regularization of the solution?

Unfortunately, the reviewer did not understand the main principal of the work, although it is clear to the other two reviewers. Element size for deformation modeling has nothing to do with the resolution of SDB. The newly added Fig. 2 should help the reader understand the principal. We have added the following sentence to its caption:

"While coseismic deformation is of long wavelength, local seafloor slope variations can lead to coherent short-wavelength features of bathymetry decrease or increase."

We have also re-arranged the text to say the following earlier: "We add the model-predicted three-component coseismic displacements to the pre-earthquake bathymetry to produce Synthetic Differential Bathymetry (SDB), in the manner illustrated in Fig. 2."

In the supplement a brief statement about the rheological parameters is given. Is it true that you do not use any lateral nor changes with depth in the Poisson's ratio or rock density? When not true please show it.

This is clearly stated in "Deformation model", now in the Methods section. No further clarification is needed.

From Supp. Figure 9, I do not see that the ODB data is consistent with the geodetic

seafloor measurements. Already the same statement by Fujiwara et al. I found weak. Please compare the deformation rates for the closest points between the profiles and the geodetic measurements.

As clearly explained in the caption of this figure (now Supplementary Fig. 6), this slip distribution “is not obtained by inversion but is based on hand-extrapolating the slip distribution shown in Fig. 3a and Supplementary Fig. 5a”. The caption goes on to say: “The purpose is not to fit all the geodetic data, but to show that the magnitude of seafloor displacements is consistent with most data, especially the ODB data at site TJT1. A more complete understanding of the heterogeneous shallow slip distribution would require more near-trench observations.”

In general, when providing plots which show data outside the range of the colour bar please also indicate the end colours (see Figure 2d). In Figure 2b the colour code white means 0 and probably also values which are outside the colour range. This is quite misleading.

Thanks for the suggestion. We have modified color bars in all relevant figures.

D. Appropriate use of statistics and treatment of uncertainties

Including ODB data in the inverse modelling would address a need for realistic error estimates of the data. Fujiwara et al. did not discuss these issues. I would like the Authors to contribute to this problem and to present an error assessment of the differential ODB data. Additional problems may rise which may introduce systematics for larger depths e.g. also given by the DOT (Dynamic Ocean Topography) which may be strongly present due to the Kuroshio current (Fuchs et al.).

We disagree. Had the pre-seismic surveys been conducted with a future trench-breaching earthquake in mind, had the water of the Japan Trench been shallower by a few km, had the Tohoku-oki earthquake occurred in the calm tropical sea, had JAMSTEC had more time to prepare for the postseismic survey, and had the technology been much better than what we have now, better data would have been obtained, and more rigorous error analyses would have been possible. See reply to the “A cross-track curvature ...” comment above.

The paper claims in the Abstract a proof of concept (dynamic weakening/strengthening). Can you give a level of confidence for this statement? Moreover, I miss an error estimate of your slip considering the error propagation of your measurements in the inverse modelling. Without this level of confidence, the statement of slip and the conclusions will always be vague.

We have changed the last sentence of the Abstract to “The determined slip distribution indicates that dramatic net weakening or strengthening of the shallow fault did not occur during the Tohoku-oki earthquake.” The newly added Figs. 6 and 7 (from previous Supplementary Information) make it very clear that the rejection of

the green and red curves in now Fig. 8 is based on tests and error analyses. We have also explained the whole matter more clearly in the newly added section “Slip gradient” under Results.

Figure 1: I think it is not appropriate to compare all the different slip-models, since the models are based on different fault geometries, processing parameters like regularization, spatial scales, all based on various observation types where each different observation type may also show different spatial sensitivity and spatial resolution. I recommend to erase Figure 1 (or to put it in the appendix). Anyhow the different models do not contribute to the final conclusion made.

The purpose of Fig. 1 is to show our current state of knowledge about shallow fault slip in this earthquake. We think it is important to keep this figure. However, to address the reviewer’s concerns, we have revised the text as follows:

“The differences are due partly to various simplifications in inverting coseismic observations to determine fault slip. For example, many of the finite fault models, especially those used for inverting tsunami data, assume a planar fault and/or consist of rather large subfaults of rectangular shape¹². Depending on how fault slip is constrained at the trench, the peak slip determined by the inversion may be located at the trench or some distance away from the trench. However, the primary reason for the poor state of knowledge is the lack of near-field observations of horizontal seafloor displacements ...”

E. Conclusions: robustness, validity, reliability

I have severe doubts that high resolution ODB data, which only show a small fraction of the trench, can provide a proof of concept extrapolated on to a distributed fault slip model. The large RMS fit and the weak improvements searching the model space may emphasize this assumption. The Authors try to formulate an overall conclusion which is from my point of view not evidently given by the dataset.

We disagree. See replies to various comments above, especially “I am a little bit surprised ...”, “A cross-track curvature ...”, and “Including ODB data in the inverse modelling ...”.

F. Suggested improvements: experiments, data for possible revision

In your model approach the reference of your deformation is set on the whole area easterly of the trench. The ODB data would offer the very unique information about deformation very close and on the other side of the trench which never has been measured before and never (at least what I know) derived in any distributed fault-slip model. I think it would be of eminent importance to include this information in your modelling, choosing areas not affected by deformation as reference. This would incredibly increase the innovative potential of the manuscript and in addition highlight the potential of ODB-Data.

We disagree. (1) The deformation field is independent of reference frame and is arbitrary to a rigid-body translation. Physically, it makes absolutely no difference what area is used as the reference. The choice is a matter of practicality, and the area seaward of the trench is the most practical in our case. (2) Given the data we have, we do not know how to choose “areas not affected by deformation as reference”.

I suggest the Authors to compute a model with and without the ODB data and to directly show the impact of ODB data on the inverse modelling.

Moreover, an inversion with other datatypes like seafloor geodetic measurements and tsunami data is most important to evaluate the different data quality aspects.

We do not know what the reviewer is suggesting. We are comparing SDB with ODB. By “compute a model ... without the ODB data”, we hope the reviewer does not mean that we should invert other types of data to determine the fault slip. There are already 45 of those models, all shown in Fig. 1.

H. Clarity and context: lucidity of abstract/summary, appropriateness of abstract, introduction and conclusions

The paper shows the methods and analysis applied, all written in a comprehensive and logical textual style with illustrative Figures and Plots.

Thanks.

Reviewer #2 (Remarks to the Author):

It is very kind of the reviewer to have sent an apologetic follow-up note to the editor about misquoting the paper as “Wang et al” instead of “Sun et al”. No problem at all. As the corresponding author, Wang should take the responsibility.

Wang et al evaluate the updip deformation caused by the 2011 Mw9.0 Japan megathrust earthquake looking at the difference between bathymetry profiles acquired before (mainly from year 1999) and after the rupture (year 2011). These data, already analysed in 2011 by Fujiwara et al. (2011), provided the first clear evidence that the megathrust rupture did reach the surface. Before the Tohoku earthquake, it had long been speculated that megathrust ruptures could not propagate through the shallow unconsolidated sediments. These bathymetry data proved otherwise and even demonstrated that the rupture can propagate to the surface with very large slip amplitudes (~60m). The aim of the Fujiwara et al. (2011) study was to release the result quickly. It was published as a Science brevia and therefore didn't go into the details of the analysis nor proposed a full discussion of the results. In this respect, the study proposed by Wang et al. addresses this lack and is very valuable.

The main scientific question posed by Wang et al. is whether a finer processing allows to distinguish a trend in coseismic slip amplitude (increase, stabilization or a decrease)

towards the trench. This piece of information is critical to constrain the frictional properties of the shallow fault interface and its ability to accumulate stresses and/or release them during a coseismic rupture. The question is therefore of fundamental interest.

The manuscript is easy to read and contains appropriate references. Yet, I'm not convinced by the demonstration and the argumentation. Although I consider that this work represent a significant contribution, I feel that my concerns would be better address by a re-submission.

I list below my main concerns:

***despite significant efforts (finite-element modelling, grid search of optimal parameters, etc), the result of the analysis is not significantly different from the more simple and straightforward analysis of Fujiwara et al. (2011). Authors state that their new analysis allows to discriminate between scenarii of slip amplitude significantly changing toward the trench, but this was already suggested by the 2011 results: the residual of the differential bathymetry didn't show any clear gradient in the residuals. The previous study was aimed at quickly releasing the results and didn't push the interpretation: in that respect the present study brings a clear plus. The discussion is of general interest and needs to be published.

Thanks for the encouragement. The method used in Fujiwara et al. (2011) assumes rigid-body translation of the forearc wedge relative to the incoming plate. It did not have any internal deformation and could not allow a slip gradient. The newly added section "Slip gradient" should make this clearer. Plus, the remnant depth bias was not determined in the past.

***several other bathymetry datasets are analysed in the supplementary section. These analyses reveal variable results, sometimes contradicting the main conclusion of the paper. This part is not sufficiently detailed considering that it shows results not in line with the main text. Two cases in particular:

*the 2004-2011 differential bathymetry (the main text is based on 1999-2011 profiles) is assumed unreliable because of a limited coverage seaward of the trench (assumed stable and used as a reference to calibrate the analysis). There is no clear explanation as to why 10km of seaward data is not sufficient to calibrate a 40km section: the residuals are small and do not show any suspicious trend (Fig. S7). This result is quite different from the 1999-2011 analysis: 12m decrease in slip for 2004-2011 versus 5m increase for 1999-2011.

We are sorry that the text was poorly worded and gave the wrong impression that the 2004 data should be rejected. We have thus revised the text as follows.

"The very short length of the seaward portion of the survey corridor, being only 1/5 of the 1999 survey, causes great difficult in using it as the reference for ODB. Therefore the ODB based on the 2004 data is considered less reliable."

*The authors also present in the supplementary section differential bathymetry results from a profile further north (Fig. S8). The results show a significant 20m increase of slip towards the trench suggesting that the shallow megathrust underwent coseismic weakening. This result suggest that there is not one single type of frictional response as proposed in the main text, but some along-strike variability.

We agree that along-strike variations in the slip value and fault behavior should be further emphasized. We have thus changed the text in many places to clarify that the results are only for the main corridor in the main rupture area. We have also emphasized in the new Discussion section that the northern corridor shows a different slip behavior.

***The authors motivate their analysis stating that finite-fault slip models of the Tohoku earthquake did not manage to resolve slip on the shallowest part of the subduction interface. I think this discussion is, for the most part, too long and not constructive:

*most finite-fault slip model listed do not pretend to resolve slip at the scale of the bathymetry profiles. For instance, Fujii et al. (2011) resolve slip on squares of 25 and 50km, bigger than the length of the bathymetry profile. Most studies listed are based on a planar fault geometry which usually fall km, sometimes more than 10, away from the trench.

We have revised the relevant text; see reply to Review #1's comment "Figure 1: I think ...". We leave more detailed discussions to a separate paper that we are writing. Also, to more faithfully illustrate the state of knowledge, we have changed "the most trenchward 50-60 km of the fault" to "the most trenchward 100 km of the fault".

*recent studies have highlighted the need to simultaneously incorporate tsunami and the 3D slab geometry to properly infer the details of the shallow slip (Bletery et al., 2014; Melgar and Bock, 2015). Even the models doing so have a spatial resolution of several km along strike, and show relatively strong variability near the trench. A fair and meaningful comparison should be limited to these recent studies and acknowledge their resolution and variability.

The final statement that "contrary to a common belief, tsunami observations do not necessarily provide resolution in determining near-trench slip" needs to be adjusted. Simple synthetic tests (e.g. Yokota et al., 2011; Romano et al., 2012; Bletery et al., 2014) reveal that tsunami data should be very good at resolving near-trench slip. The statement made by the authors could be true if tsunami data were not modelled properly (e.g. over-simplified fault geometry as in many studies) or if the authors are talking about length scales smaller than what is resolved by tsunami data (10km?). In the latter case, it is not obvious that differential bathymetry is doing better.

We have deleted the sentence "Contrary to common expectations, the use of tsunami data so far has not improved the situation." In the caption for Supplementary Figure

1, we have revised the final sentence as: “The use of tsunami data in some of the recent models helped improve near-trench resolution of slip models such as models 26 and 35, but not in all the recent models.” Here 26 and 35 are the models of Bletery et al. (2014) and Melgar and Bock (2015), respectively. We think it is pertinent to point out that the improvement is not in all the recent models (e.g., Minson et al., 2014; Romano et al., 2014; Wei et al., 2014; etc.).

*** The argument that "Large afterslip of the shallow megathrust [...] is also extremely unlikely, given the absence of interplate aftershocks along the main rupture zone" is not supported by any observation. Indeed, several recent postseismic studies (Perfettini and Avouac, 2014; Yamagiwa et al., 2015) argue that the shallowest part of the megathrust

We have added a sentence: “Post-seismic seafloor GPS measurements do not indicate afterslip in this area, although they do suggest large near-trench afterslip to the south of the main rupture zone (Sun and Wang, 2015).” The shallow afterslip discussed in Yamagiwa et al. (2015) is south of the main rupture area and is similar to the results in Sun and Wang (2015). However, the large shallow afterslip in Perfettini and Avouac (2014) contradicts seafloor GPS observations.

Additional comments and corrections:

_Figure 1 and S1. Romano et al. (2012), line 15, have a maximum amplitude of more than 30m (over 40m?). I leave the authors the responsibility to double check all profiles.

We use the more recent result in Romano et al. (2014). All profiles have been double checked.

_a discussion of the shortening rates and the limit of elasticity could be a nice addition.

By “shortening rate”, perhaps the reviewer means strain, as the model has no time dependence. The preferred model shows rather low tensile strain at long wavelengths because of the small slip gradient. Elastic limit may (or may not) be exceeded at short wavelengths. We have revised the text as follows.

“Some of the differences between the optimal SDB and the ODB, especially in the amplitude of the short-wavelength features near the trench, may be due to inelastic deformation during or shortly after the earthquake¹⁵ that are not modeled in this work. They also contribute to the relatively large RMS.”

_a direct comparison with the result of Ito et al. (2011)

We do not feel very comfortable to single out Ito et al. (2011) to criticize. The following is solely for the review’s and editor’s information. We are very familiar with Ito et al.’s (2011) results and have discussed some of the issues with Dr. Ito privately. The acoustic ranging measurements in that work have very large uncertainties. Even for the same location (TJT2), two measurements yielded

displacements differing by 30–40 m. The vertical displacement recorded at TJT1 has a much smaller error range (0.5 m), and we thus compare our model to this observation (Supplementary Figure 6, new numbering). The fault dip of 3° used by Ito et al. (2011) was too small and contributed to an overestimate of near-trench slip (80 m).

_data to reproduce the results are not included. The authors comment in their manuscript that no finite-fault source inversion study has included the results of the previous differential bathymetry analysis (Fujiwara et al., 2011): for this not to happen again, I urge the authors to provide their new results in ascii format in the supplements (coordinates of the profiles, slip amplitudes inferred).

We have now added end coordinates of the ODB profiles to relevant figures. The slip values for each profile (both preferred and testing) are very simple and are shown in the figures. The model in Supplementary Figure 6 (new numbering) is for reality check, to show that the SDB modeling does not cause trouble to fitting other geodetic data. Although it shows amazing agreement with the GPS measurements (none were used to constrain the model), it is not THE final model that we would want to promote.

Reviewer #3 (Remarks to the Author):

One of the most important lessons of the great 2011 Tohoku-oki earthquake is the unexpectedly large slip that seems to have occurred at shallow depth towards the trench. In the classical literature, the area between the trench and some up-dip limit of the seismic friction plane was suspected to be "weak" and creeping, in relation with the expected large amount of water carried by sediments at décollement-level.

This paper specifically addresses the rate of change of slip towards the trench axis, using bathymetric data that have been collected before and just after the earthquake. The data were already presented shortly after the event, but their exploration was never pushed to the level presented here. Beyond the general problem of the amount of co-seismic sliding towards the trench, the aim here is to reach the rate of change of this slip, since this value is expected to change drastically with mechanical behaviour. This approach is definitely novel.

The paper is very well written and presented, and I do not have much to say about the overall writings, illustrations, and references. One detail in fig 4b: curves labelled 1 and 2 (blue) are described in the text only, it could be useful to add a short text in the legend of the figure as well, and in the meantime, indicate that 2 is your favoured scenario.

We have added the following sentence to the caption of Fig. 8 (previous Fig. 4):

“Blue curve 2 represents a more likely scenario in which delayed dynamic weakening (Noda and Lapusta, 2012; Smith et al., 2015) occurred on the shallow

fault during the earthquake.”

Here are a few comments (dealing mainly with minor precisions on the methodology):

- Figure 1 is impressive, and the authors made a nice and very useful job in compiling these various models. There are clearly two classes of models in the 0 to 80 km range from the trench: some of the models show a bell curve (including or not including seafloor geodesy), while some others (few) show steadily rate increase. Is there any "rule" behind this (type of inversion used), or is it that the behaviour towards the trench is just not constrained in most cases?

The revised text fully addresses the reviewer’s question; see reply to Review #1’s comment “Figure 1: I think ...”.

- Some of the models shown use a Bayesian approach, and it may be fair to state that in that case, the inversion does not lead to one single slip model but to a population of models. If the Bayesian approach is correct, then the solution proposed in this manuscript should belong to this population. I understand that it would be hard to check.

Based on our own understanding and experience (the second author K. Wang did his PhD on Bayesian inversion), not all Bayesian models need to be presented as populations of models, and non-Bayesian models can also be presented as populations of models.

- The critical range used here covers the first 40 km of the wedge away from the trench. In Fujiwara et al. 2011 (using these spectacular bathymetric data for the first time), the area near the trench axis was excluded in relation with possible landslides. Perhaps a word should be added on the possible limitation that co-seismic sediment mobilization would produce. I guess that the area mentioned as landslide in their figure 1 has been excluded from the calculation (shows also in your figure 2d).

As the reviewer correctly said, that area is indeed excluded from our calculation. We have modified Figure 3 (previous Fig. 2) caption to further clarify this.

- The trade-off between the average slip and the depth adjustment parameter is well explained as an acoustic bias. Theoretically, it would be possible to link the final value of this depth adjustment to the kind of seasonal variations that are expected to be found in the water column, although it is most probably within the error bars of anything that could be calculated. Practically, there might be a way to check the water velocity law variation in the trench area, if a flat area exists: beams with larger aperture will wrongly bend upward or downward if the velocity law is incorrect, and comparing the required velocity laws before and after would ultimately lead to the adjustment. At least the sign of it may be retrieved ... not for this paper anyway!

Good suggestion for future work. The best way is to conduct full 3D time-dependent sound speed tomography based on multiple sensors. We are very optimistic that this

will happen soon, as the technology is already there.

- The final model proposed here is finally a sort of upper-bounding envelope of the models proposed so far. If this applies not only to this section, but also to the entire sliding area, would not the modelled magnitude of the earthquake be too large compared to the observed one?

The amount of slip in our optimal model (now Fig. 8) is indeed larger than in most, although not all, published models along the specific track. We expect the fault slip to vary along strike (Supplementary Figure 6). We have added to the caption of Supplementary Figure 6: “This model represents an earthquake of $M_w=9.02$ if rigidity is assumed to be 40 GPa.” We also added a larger-area view of this model to this figure to show its compatibility with land-based GPS observations.

My conclusion is that the paper is based on high-quality data, the methodology is robust and very well explained. The paper brings a new and important piece to the understanding of the co-seismic mechanical behavior during megathrust earthquakes.

Thanks for the encouragement.

Reviewer #1 (Remarks to the Author):

Based on the changes made to the manuscript, I agree to publish the paper in Nature-Geoscience after including two issues (as stated below).

- (1) Include an error bound of the accuracy of the measurements.
- (2) Give a level of confidence to the derived slip distribution.

regards, M. Fuchs

Responses to Reviewers' Comments

Reviewer #1 (Remarks to the Author):

A. Summary of the key results

The presented manuscript "Large fault slip peaking at trench in the 2011 Tohoku-oki earthquake" presented by Sun et al. 2016, addresses and discusses a new data type namely "measured ocean acoustic profiles" in the use of inverse modelling. ODB data thereby is a very important datatype since ocean topography offers here information on the co-seismic deformation close to the trench where almost no direct observations are available.

B. Originality and interest: if not novel, please give references

The originality of the paper is limited because the manuscript does not clearly focus on the contribution of ODB data in inverse modelling. Fujiwara et al. already discussed all data related topics and drew conclusions on the near trench deformation. In addition, various other Authors have already shown that there has been peaking slip close to the trench for the 2011 Tohoku-Oki earthquake (see e.g. Hooper et al.). I see the scientific significance of the paper limited and do not recommend to publish the presented manuscript without major changes and severe improvements in the ODB data quality.

>We respectfully disagree with the reviewer's opinion on the originality of this work.

>As is obvious to the other two reviewers, our work is clearly focused on the

>contribution of ODB data to deciphering coseismic slip of the shallow part of the

>megathrust and has provided new and quantitative information.

>> I still have doubts on the presented data quality and its practical use in distributed slip modeling.

>> However, with the focus of the paper as representative example, ODB is highlighted as

>> an important additional information source and therefore may encourage future work in this direction.

C. Data & methodology: validity of approach, quality of data, quality of presentation

The derived ODB profiles are used as input quantity for the inverse modelling but lack from a bias which may be present in the data coming from variations in water temperature over depths. The relation of acoustic parameters like pressure, temperature, saltness etc. may not follow a linear relation to the measured profiles since already the geometry of the profiles follow a $1/\cos(\phi)$ relation. So the estimation of a single bias for the whole profile may be a weak simplification.

>We did not make it clear that in processing the bathymetry data, we already

>considered the seasonal variations in water temperature over depths. For surveys at
>different times, Fujiwara et al. (2011) used different temperature-depth profiles to
>derive water sound speeds and the seafloor depths. We have revised the text as
>follows to clarify this:
>"Although temporal and spatial variations in water temperature, especially at shallow
>depths (< 2000 m below sea surface), have been accounted for in deriving sound
>speed structure of ocean water for ODB determination, remaining uncertainties still
>lead to some remnant depth bias in the bathymetry data, even after maximizing cross
>correlations of the seaward (reference) side of different surveys.

>> Changes accepted

"

Do the residuals meet the criteria of Gaussian white noise? What noise properties characterize the ODB in the inverse modelling?

If we correctly get the message behind these questions, they are asking whether it is appropriate to use RMS (based on the L2-norm) as a measure of SDB – OBD differences. We actually had tested the mean absolute deviations (based on the L1-norm, much more tolerant to "outliers") and obtained exactly the same optimal parameter values as with the RMS. This shows that the results are not sensitively dependent on the type of "noise". We have clarified this by adding the following sentence after the first mentioning of RMS:

"We have also done the search by minimizing the mean absolute deviation of SDB from OBD and obtained the same results as with the RMS."

>> Changes accepted

I am a little bit surprised by the large RMS fit of ~8.5 meter which I personally find poor. Looking at the maximal value of ~25meter and the distribution of most of the data below ~10 meter in amplitude reveals a high uncertainty in the data or model. Moreover, the difference in the biases for different reference ODB is also quite high (~6m). From the point of accurate seafloor modelling I would propose at least meter accuracy which is in agreement with DeSanto et al.. De Santo showed that the data of Fujiwara et al. have an accuracy of around 20m which may not be appropriate for analysis.

>The reviewer is misled in this regard, perhaps because we did not explain it
>adequately. We have added Fig. 2 (from previous Supplementary Information) to
>help the reader understand the issue. We have also added the following text right
>after the above newly added sentence:
>"Incoherent short-wavelength fluctuations in the ODB associated with sea and
>seafloor conditions, stability of the acoustic and navigation systems, and errors in
>local water temperature and salinity profiles are not minimized, partly responsible
>for the relatively large RMS. For our study, the useful information is from longwavelength
>co-seismic deformation and coherent short-wavelength differential
>bathymetry due to topographic shift as shown in Fig. 2. The useful information is
>reflected in the RMS differences between different models that are based on the
>same data set."

>> Changes accepted

We do not understand the reviewer's comment about differences in depth biases between different models being large ("high"). We searched a wide range of depth bias values. Of course the values are very different between models.

>> I do not mean the chosen search space nor the applied bias. As shown above an accuracy of
>> 1m should be obtained depending on the resolution chosen.
>> The bias is thereby not the main driver but may lead to long wavelength uncertainties over
>> the entire scene. If you obtain large differences for the depth Bias I am not sure how this
>> affects your results.

A cross-track curvature of the measured deformation profiles at the deep sea terrace (Figure 2d) is clearly visible. It is very likely that the data is corrupted by instrumental errors. The model data (Figure 2c) show here flat terrain which seems appropriate.

>We already pointed out this issue in the figure caption (and shaded that area) which may have escaped the reviewer's attention.

>Because of the large uncertainties, we
>excluded this area from RMS calculation. Our effort is devoted to overcoming such
>difficulties to extract valuable information from these data. We hope the review will
>agree with us that we should not "throw the baby out with the bath water."

>> Comments accepted

The search in the parameter space Figure 3 shows only a small improvement of the model fit 8.9 to 8.5 (~5% to "worst case"). Probably your "simulated" parameters are not very efficient. It seems that the model is not capable to model the fine scale ODB information properly.

>The percentage comparison (5%) is logically incorrect; see reply to the "I am a little
>bit surprised ..." comment above. However, the reviewer's comment does remind us
>that the color bar in the original figure is confusing. We have modify it to show that
>the actual upper-bound is much higher than 8.9. We have also added the following
>sentence to Fig. 4 (previous Fig. 3) caption:
>"The maximum RMS deviation is 12.5 m (for 40-m slip and 0-m depth adjustment),
>but the color scale saturates at 8.9 m."

>> Change accepted

The acoustic profiles offer an approximately resolution at 4km depth of ~150m and the model a resolution of around 5km (what I can determine from the supplementary of Figure 3). How is this compatible? How does the mode treat the high resolution of the data and the involved spatial regularization of the solution?

>Unfortunately, the reviewer did not understand the main principal of the work,
>although it is clear to the other two reviewers. Element size for deformation
>modeling has nothing to do with the resolution of SDB. The newly added Fig. 2
>should help the reader understand the principal. We have added the following
>sentence to its caption:
>"While coseismic deformation is of long wavelength, local seafloor slope variations
>can lead to coherent short-wavelength features of bathymetry decrease or increase."
>We have also re-arranged the text to say the following earlier: "We add the model predicted
>>three-component coseismic displacements to the pre-earthquake
>bathymetry to produce Synthetic Differential Bathymetry (SDB), in the manner
>illustrated in Fig. 2."

>> Changes accepted

In the supplement a brief statement about the rheological parameters is given. Is it true

that you do not use any lateral nor changes with depth in the poissons ratio or rock density? When not true please show it.

>This is clearly stated in "Deformation model", now in the Methods section. No
>further clarification is needed.

>> This circumstance could be responsible for severe model errors. Plenty of papers treated this issue.

>> Please clarify in some sentences the introduced errors of this simplification.

From Supp. Figure 9, I do not see that the ODB data is consistent with the geodetic seafloor measurements. Already the same statement by Fujiwara et al. I found weak. Please compare the deformation rates for the closest points between the profiles and the geodetic measurements.

>As clearly explained in the caption of this figure (now Supplementary Fig. 6), this
>slip distribution "is not obtained by inversion but is based on hand-extrapolating the
>slip distribution shown in Fig. 3a and Supplementary Fig. 5a". The caption goes on
>to say: "The purpose is not to fit all the geodetic data, but to show that the magnitude
>of seafloor displacements is consistent with most data, especially the ODB data at
>site TJT1. A more complete understanding of the heterogeneous shallow slip
>distribution would require more near-trench observations."

>> Please state also in the text: the slip distribution is obtained by hand-extrapolating the
>> slip distribution shown in Fig. 3a and Supplementary Fig. 5a

In general, when providing plots which show data outside the range of the colour bar please also indicate the end colours (see Figure 2d). In Figure 2b the colour code white means 0 and probably also values which are outside the colour range. This is quite misleading.

>Thanks for the suggestion. We have modified color bars in all relevant figures.

>> Changes accepted

D. Appropriate use of statistics and treatment of uncertainties

Including ODB data in the inverse modelling would address a need for realistic error estimates of the data. Fujiwara et al. did not discuss these issues. I would like the Authors to contribute to this problem and to present an error assessment of the differential ODB data. Additional problems may rise which may introduce systematics for larger depths e.g. also given by the DOT (Dynamic Ocean Topography) which may be strongly present due to the Kuroshio current (Fuchs et al.).

>We disagree. Had the pre-seismic surveys been conducted with a future trenchbreaching
>earthquake in mind, had the water of the Japan Trench been shallower by a
>few km, had the Tohoku-oki earthquake occurred in the calm tropical sea, had
>JAMSTEC had more time to prepare for the postseismic survey, and had the
>technology been much better than what we have now, better data would have been
>obtained, and more rigorous error analyses would have been possible. See reply to
>the "A cross-track curvature ..." comment above.

>> I agree that the data quality can not be improved afterwards. But for the interested reader
>> the key issue is how accurate are the measurements. I think it is not appropriate
>> that you find improvements on the slip distribution without giving any value of

>> confidence (see comments below). Please give an error estimate concerning your measurements and
>> your derived slip distribution.

The paper claims in the Abstract a proof of concept (dynamic weakening/strengthening). Can you give a level of confidence for this statement? Moreover, I miss an error estimate of your slip considering the error propagation of your measurements in the inverse modelling. Without this level of confidence, the statement of slip and the conclusions will always be vague.

>We have changed the last sentence of the Abstract to "The determined slip distribution indicates that dramatic net weakening or strengthening of the shallow fault did not occur during the Tohoku-oki earthquake." The newly added Figs. 6 and 7 (from previous Supplementary Information) make it very clear that the rejection of the green and red curves in now Fig. 8 is based on tests and error analyses. We have also explained the whole matter more clearly in the newly added section "Slip gradient" under Results.

>> Changes accepted

Figure 1: I think it is not appropriate to compare all the different slip-models, since the models are based on different fault geometries, processing parameters like regularization, spatial scales, all based on various observation types where each different observation type may also show different spatial sensitivity and spatial resolution. I recommend to erase Figure 1 (or to put it in the appendix). Anyhow the different models do not contribute to the final conclusion made.

>The purpose of Fig. 1 is to show our current state of knowledge about shallow fault slip in this earthquake.

>> I disagree that the slip distributions do represent our current knowledge. The distributions are represent
>> snapshots for different datatypes based on different modeling. Shallow slip has already been discussed in
>> the literature.
>> The plenty models can be concluded in some sentences with reference.
>> Moreover, the various models presented do no have any impact on your derived results.

>We think it is important to keep this figure. However, to
>address the reviewer's concerns, we have revised the text as follows:
>"The differences are due partly to various simplifications in inverting coseismic
>observations to determine fault slip. For example, many of the finite fault models,
>especially those used for inverting tsunami data, assume a planar fault and/or consist
>of rather large subfaults of rectangular shape¹². Depending on how fault slip is
>constrained at the trench, the peak slip determined by the inversion may be located at
>the trench or some distance away from the trench. However, the primary reason for
>the poor state of knowledge is the lack of near-field observations of horizontal
>seafloor displacements ..."

>> Changes accepted

E. Conclusions: robustness, validity, reliability

I have severe doubts that high resolution ODB data, which only show a small fraction of the trench, can provide a proof of concept extrapolated on to a distributed fault slip

model. The large RMS fit and the weak improvements searching the model space may emphasize this assumption. The Authors try to formulate an overall conclusion which is from my point of view not evidently given by the dataset.

>We disagree. See replies to various comments above, especially "I am a little bit surprised ...", "A cross-track curvature ...", and "Including ODB data in the inverse modelling ...".

>F. Suggested improvements: experiments, data for possible revision

>> Changes accepted

In your model approach the reference of your deformation is set on the whole area easterly of the trench. The ODB data would offer the very unique information about deformation very close and on the other side of the trench which never has been measured before and never (at least what I know) derived in any distributed fault-slip model. I think it would be of eminent importance to include this information in your modelling, choosing areas not affected by deformation as reference. This would incredibly increase the innovative potential of the manuscript and in addition highlight the potential of ODB-Data.

We disagree. (1) The deformation field is independent of reference frame and is arbitrary to a rigid-body translation. Physically, it makes absolutely no difference what area is used as the reference. The choice is a matter of practicality, and the area seaward of the trench is the most practical in our case. (2) Given the data we have, we do not know how to choose "areas not affected by deformation as reference".

>> Estimating distributed slip on the other side of the trench would highly improve
>> the innovative potential of the paper. However, based on the ODB data alone it may
>> not be applicable. Response accepted.

I suggest the Authors to compute a model with and without the ODB data and to directly show the impact of ODB data on the inverse modelling.

Moreover, an inversion with other datatypes like seafloor geodetic measurements and tsunami data is most important to evaluate the different data quality aspects.

We do not know what the reviewer is suggesting. We are comparing SDB with ODB. By "compute a model ... without the ODB data", we hope the reviewer does not mean that we should invert other types of data to determine the fault slip. There are already 45 of those models, all shown in Fig. 1.

>> Response accepted

H. Clarity and context: lucidity of abstract/summary, appropriateness of abstract, introduction and conclusions

The paper shows the methods and analysis applied, all written in a comprehensive and logical textual style with illustrative Figures and Plots.

Thanks.

Reviewer #2 (Remarks to the Author):

I think the authors have done a substantial work to improve the manuscript and address most of the concerns raised by the reviewers.

I would just ask one modification in the manuscript:

L105-107 : it's the modelling of GPS which indicates no afterslip, not just the GPS data. You could rewrite the sentence as :

"Modelling of post-seismic seafloor GPS measurements does not indicate afterslip in this area, although it does suggest large near-trench afterslip to the south of the main rupture zone."

Reviewer #3 (Remarks to the Author):

The new version is ready for publication, from my point of view:

- Questions regarding figure 1 have been fully answered. I do follow the authors in keeping this figure in the main text, and not rejecting it to the supplementary material. This compilation summarizes the state of knowledge on the distribution of slip, and it is now clearly stated in the text that the underlying assumptions beneath each of these models are variable. It may sound annoying for some of the scientists who produced some of these models, since there may be plenty of good (and bad) reasons for the observed scattering in the slip distribution and peak of slip close to the trench. This includes the fact that some of these models did not try to reach the level of resolution that is searched here. However, slip distribution remains one major output of these models, and comparing the solutions is relevant and should not be removed.

- Moving figure 2 from the supplementary to the main text is useful, since there seemed to be some misunderstanding on the fundamental basis of the method. Questions regarding the reference frame obviously relate to this. Something that I could not find is a statement on the non-use of the trenchward side in the calculation of the RMS. A remote question, for curiosity, is the change of sign of the ODB in the trench between port and starboard sides for the main corridor, not occurring for the northern corridor. Is it related to some uncorrected "attitude" of the vessel?

- The new section on the "slip gradient " is fine, and figure 5 together with figures 6 and 7 that were taken from the old supplementary material make the discussion on that matter concise.

- Added comments on the northern section and the variability of the slip behavior are useful. Geodetic model in supplementary material answers both the questions of along-strike variability and size of the earthquake. One has to admit that the main corridor was right at the peak of slip, or close to it, but I do understand that nothing more can be done on the along-strike variability at least using the ODB.

More generally, the lesson (to me at least) is that remote data alone cannot solve for the full complexity of the slip distribution, and in situ data (such as bathymetry, seismic, ocean floor stations,...) are needed. The supplementary figure 1 for models that include tsunami data as constraints is a clear demonstration that despite improvement, resolution is still limited near the trench. Regarding land-stations, the loss of resolution is clearly related to the distance to the trench, and this has long been recognized as a limiting factor for many subductions (both for interseismic coupling and coseismic slip), perhaps with the exception of the Chile and Makran subductions. Although the technique itself is not simple (vessels are not satellites), and the source of errors are still to be elucidated and quantified, the potentiality of differential bathymetry is very high. In that regard, this paper is a very important stone.

Line 67: indicating ?

Responses to Reviewers' Final Comments

Reviewer #1:

The reviewer essentially accepted all the changes we made in response to his original comments. Answers to the few new comments are as follows.

Based on the changes made to the manuscript, I agree to publish the paper in Nature-Geoscience after including two issues (as stated below).

- (1) Include an error bound of the accuracy of the measurements.
- (2) Give a level of confidence to the derived slip distribution.

To address point (1), we have added the following paragraph to the **Differential bathymetry before and after the earthquake** subsection under **Results**: “Because seafloor displacement between 1999 and 2004 is expected to be very small, if not negligible, bathymetry differences between 1999 and 2004 provide an error estimate for ODB. The error thus estimated by Fujiwara et al.¹ in terms of inferred total horizontal seafloor displacement is about 20 m, or about 10 m if resolved to the trench normal direction. This error to a large part is due to the inadequate length of the seaward section of the 2004 data. When applied to the 1999–2011 ODB, the actual error should be much less but difficult to quantify. Nonetheless, we do not expect the 1999–2011 error in the trench-normal direction to be much larger than 5 m.”

To address point (2), we have added the following sentence after the first sentence of **Discussion**: “Uncertainties in this slip distribution are reflected in the sensitivity plots of Figs. 4 and 5.”

Regarding our use of uniform materials properties such as Poisson's ratio or rock density, the reviewer has the following new comment: “This circumstance could be responsible for severe model errors. Plenty of papers treated this issue. Please clarify in some sentences the introduced errors of this simplification.”

What the reviewer said here may be relevant to deformation away from the rupture zone (but not “severe”). For our study area, this is not an issue at all. To clarify this, we have rewritten the relevant sentences in **Methods** as follows: “It can be readily shown that given slip distribution, the effect of spatial variations in rocks' mechanical properties on affecting elastic coseismic deformation directly above the thrust fault is negligibly small, although the effect can be larger for deformation farther away or if stress drop instead of slip distribution is prescribed to the fault. Therefore, we use uniform values for the rigidity (40 GPa), Poisson's ratio (0.25), and rock density (3300 kg m⁻³).”

Please state also in the text: the slip distribution is obtained by hand-extrapolating the slip distribution shown in Fig. 3a and Supplementary Fig. 5a.

We have now stated this in the last sentence of Discussion.

Reviewer #2 (Remarks to the Author):

I would just ask one modification in the manuscript:

L105-107: it's the modelling of GPS which indicates no afterslip, not just the GPS data.

You could rewrite the sentence as: "Modelling of post-seismic seafloor GPS measurements does not indicate afterslip in this area, although it does suggest large near-trench afterslip to the south of the main rupture zone."

We have made the suggested change.

Reviewer #3 (Remarks to the Author):

Something that I could not find is a statement on the non-use of the trenchward side in the calculation of the RMS. A remote question, for curiosity, is the change of sign of the ODB in the trench between port and starboard sides for the main corridor, not occurring for the northern corridor. Is it related to some uncorrected "attitude" of the vessel?

We have now modified Fig. 3 caption to say "... these segments *as well as that seaward of the trench* are not included for calculating the RMS deviation." Here newly added words are italicized. The port–starboard asymmetry mentioned by the reviewer seems to be related to beam angles (grazing angles) in the across-track direction. We checked the roll bias and error of the mounting angle of the transducer array but found the degree of the asymmetry vary from one situation to another. Attitudes of the vessel may relate to the cause, especially considering the fact that the asymmetry appears to be worse for larger water depths. Although the cause is not fully understood, we have confidence in using the seaward side of the ODB profile as the deformation reference because the asymmetry is much less pronounced or diminished as we move east (seaward) and away from the trench, as shown by the complete ODB profile in Fujiwara et al. (2011). The more distant part of the seaward side of the ODB profile is not reproduced in our new paper in order to keep our figures compact.

The typo in "indicating" has been fixed.